# Workplace Health Promotion, Employee Wellbeing and Loyalty during Covid-19 Pandemic—Large Scale Empirical Evidence from Hungary

**Eva Gorgenyi-Hegyes [1,*], Robert Jeyakumar Nathan [2] and Maria Fekete-Farkas [3]**

[1] Doctoral School of Economic and Regional Sciences, Hungarian University of Agricultural and Life Sciences, H-2100 Gödöllő, Hungary

[2] Academic Innovation and Product Intelligence, Multimedia University, Cyberjaya 63100, Malaysia; robert.jeyakumar@mmu.edu.my

[3] Institute of Economic Sciences, Hungarian University of Agricultural and Life Sciences, H-2100 Gödöllő, Hungary; farkasne.fekete.maria@uni-mate.hu

[*] Correspondence: gorgenyieva@gmail.com

**Abstract:** Corporate social responsibility (CSR) has become an innovative strategic management tool of socially and environmentally conscious business organizations in the 21st century. Although external CSR activities are better researched, firms' internal CSR activities such as workplace health promotion and its impact on employee wellbeing are less understood, especially during a pandemic where job security is relatively lower in many sectors of employment. Additionally, wellbeing and good health have been recognized as important targets to achieve as part of the United Nation's Sustainable Development Goal 3. Therefore, this study investigates the relationship between health-related work benefits and employee wellbeing, satisfaction and loyalty to their workplace. Large scale survey research was performed with responses from 537 employees in Hungary and 16 hypotheses were tested. Data analysis and path modelling using PLS-SEM (Partial Least Squares Structural Equation Modelling) reveal two-layers of factors that impact employee wellbeing, satisfaction and loyalty. We term this as 'internal locus of control' and 'external locus of control' variables. Internal locus of control variables such as mental and emotional health leads to wellbeing at the workplace but do not directly impact employee satisfaction and loyalty. In contrast, external locus of control factors such as healthcare support leads to wellbeing, satisfaction and loyalty. Employer commitment to healthcare support system is found pertinent especially during the pandemic. We discover wellbeing as a unique standalone construct in this study, which is vital as is it formed by mental and emotional wellbeing of employees, albeit not a determinant of employee workplace satisfaction and loyalty. We theorize workers' self-reliance and preservation as possible explanations to the disassociation between employee wellbeing and loyalty to workplace during times of crisis and the pandemic.

**Keywords:** workplace health promotion; CSR; social sustainability; PLS-SEM; self-reliance and preservation; employee wellbeing; employee satisfaction; SDG Goal 3; COVID-19

## 1. Introduction

In recent decades, the concept of sustainable development has changed significantly, in addition to the issue of the optimal use of production and resources, attention is increasingly focused on social welfare, its preservation and enhancement. The key to wellbeing is health itself, and hence, the health-conscious consumer behavior. Therefore, some of these special areas of sustainable development have become central to governmental policy decision makers in order to respond to different social and environmental problems effectively, but this need has also appeared and been discussed at corporate and consumer levels. The development of health awareness as a social sustainability factor can be implemented on the following three levels:

1. State or governmental level—through the networks within the healthcare system, health policy and measures;
2. Corporate level—through human resource management, primarily based on internal CSR activities such as workplace health promotion;
3. Consumer level (attitudes, motives, habits) (Toussaint et al. 2021).

The inclusion of sustainability at the company level is usually identified as CSR activities. CSR as an innovative part of continuously improving human resource management has become a common practice in socially and environmentally friendly business organizations. Related literature primarily identifies it with environmental awareness and environmental protection but there are many other aspects of it. External CSR activities or CSR activities (without grouping) are better researched (Hameed et al. 2016; Mihai and Bakkenist 2018), however more and more attention has recently been directed towards internal CSR activities such as workplace healthcare promotion due to the current social and economic challenges, such as labor shortages, emigration of qualified workforce and the pandemic situation such as the recent COVID-19 outbreak that caused widespread lockdowns across the world.

The amount of empirical research and results related to internal CSR—especially to the field of workplace health promotion—is limited but continuously increasing, both on national and international levels. Nevertheless, there is no doubt that the pressure on companies to engage in CSR has significantly increased (McWilliams and Siegel 2000). Therefore, the implementation of CSR activities into business practice is not only a modern phenomenon, but a necessity for all companies enhancing them towards sustainable and effective operation (Thao et al. 2019). More and more companies and managers recognize not only the necessity of social responsibility but its usefulness at the corporate level as well. A socially responsible company has advantages over its competitors with increased productivity and competitiveness, improved corporate reputation and image and significant cost savings due to the loyalty and motivation of their external and internal stakeholders. One of the most important internal stakeholders of enterprises are their employees. Employee loyalty has become a key issue of strategic human resource management; hence the focus has shifted to employee retention. With an effective retention plan, companies can reduce their long-term losses derived from the continuous fluctuation and labor shortages (Cloutier et al. 2015; Hong et al. 2012; Lee et al. 2018). While in recent years employee turnover and labor shortages have posed challenges for companies (Gorgenyi Hegyes and Farkas 2019), they now have to deal with factors that are linked to employee wellbeing. Nowadays home office, flexible working hours and work/life balance are highly appreciated by the employees because of the pandemic. Therefore, emotional health has become especially important. Nevertheless, due to the lockdown, the risk caused by COVID-19 and the compulsion of the home office, uncertainty is very high now; thus, financial benefits are presumably appreciated more than non-financial benefits as employees think in shorter term.

There is an increasing need to understand how the employer's effort to promote workplace health and wellbeing has impacted their workers' real wellbeing, satisfaction and loyalty to the employer. Following an extensive literature review, exploratory secondary research (via content analysis of corporate CSR and sustainability reports) and expert consultations (via in-depth interviews) performed by the Authors, a conceptual framework was developed that includes all the benefits and issues (as factors) related to health preservation and health promotion that can affect employee loyalty through employee satisfaction and wellbeing. A healthy and satisfied employee can turn into a loyal employee, and this results in lower labor turnover, a better corporate culture, a more cohesive community and thus better corporate performance, productivity and increased competitiveness. Furthermore, there is a real threat of migration of qualified employees, especially from Central and Eastern Europe (Bite et al. 2020). Moreover, the focus of the transforming strategic human resource management has now completely shifted from

recruitment to retaining valuable employees, therefore, this research field is also becoming more and more valuable to corporations (Lee et al. 2018).

The primary goal of this study is to identify and measure the most important health related factors and indicators which play an important role in employee wellbeing and loyalty. The research article is organized in the following structure. Introduction is followed by an extensive literature review in Section 2, synthesizing the most relevant previous seminal studies and works related to corporate social responsibility with a special focus on workplace health promotion, employee satisfaction and loyalty. Section 3 presents the research methodology where research framework is illustrated and data collection and analysis procedures are discussed. Section 4 includes relevant research findings and the results of hypotheses testing. Section 5 highlights the main conclusions of the study and emphasize its contribution to the research field not only in academic level, but also through its practical implication. In addition, recommendations for further research are presented in this final section.

## 2. Literature Review

### 2.1. Importance of Health and Health Awareness as a Social Sustainability Factor

Based on the Brundtland report, "sustainable development is a development that meets the needs of the present without compromising the ability of future generations to meet their own needs" (WCED 1987). Similar to the original concept, a significant proportion of researchers start from the production and resource use side when assessing sustainability. Nevertheless, a much more practical and detailed approach has become popular in last few years, according to which sustainability is the ability to pursue a well-defined behavior indefinitely without the deterioration of natural, human, and intellectual resources (Crittenden et al. 2011). Despite the many different definitions, it can be clearly identified that we can primarily evaluate sustainability along the well-known three basic dimensions—environmental, economic and social aspects. However, more recently, research have also been conducted from a consumption perspective. As a result, Figure 1 demonstrates we can interpret and analyze the common sections of the basic dimensions—socio-economic (e.g., job creation, skills development, business ethics, etc.), socio-environmental (e.g., health and wellbeing, global environmental change, crisis management, etc.) and eco-efficiency (e.g., life cycle management, resource management, etc.) subdimensions (ConocoPhillips Company 2006; Barcan 2016).

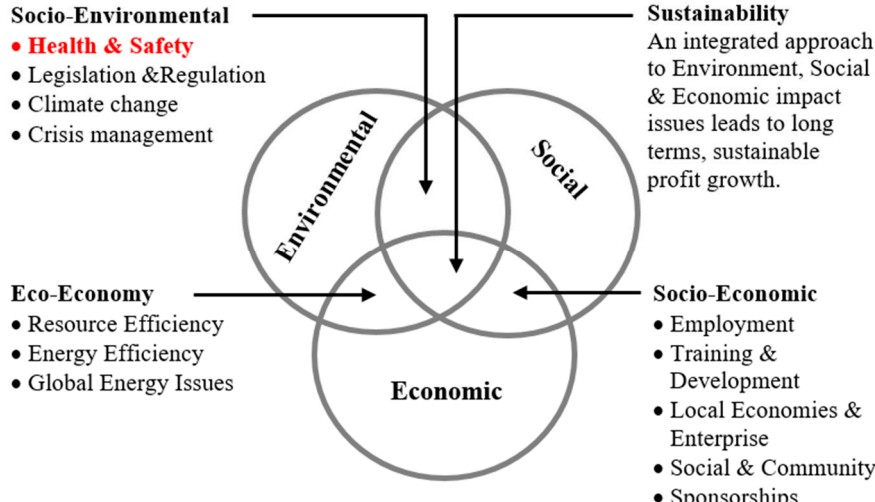

**Figure 1.** Basic pillars and subdimensions of sustainability. Source: Authors' own compilation based on ConocoPhillips ConocoPhillips Company (2006) and Barcan (2016).

Since more and more economies have recognized that our current consumer patterns and habits are no more sustainable long term and sustainable development has become a

key issue in the level of policy decision makers both in international and national level, sustainable development goals were redefined and reformulated by United Nations in 2015. Good health condition and wellbeing can be connected directly to SDG 3 (Good health and wellbeing), however indirectly to SDG 8 (Decent work and economic growth)—through satisfied and healthy employees—and SDG 12 (Responsible consumption and production)—through healthy nutrition and sustainable consumption (UN 2020).

The most commonly used and accepted concept of health was defined by the World Health Organization (WHO) in 1948. According to the Preamble to the Constitution of the WHO: "Health is a state of total physical, mental and social wellbeing and not merely the absence of disease or infirmity." (WHO 1948). The approach that maintaining and improving health requires not only scientific developments and various health services, but also the lifestyles represented by individual consumers and also the whole society, is becoming increasingly accepted. Furthermore, in the last decade WHO has placed increasing emphasis on the social and economic environmental aspect of health (McMichael 2006). Consequently, our health is affected not only by genetics and basic sociodemographic characteristics such as gender or age, but also by income and social status, educational level, cultural diversities, physical environment, working conditions, social support networks, healthcare system and personal behavior (WHO 2017). Several previous studies present relations between individual behavioral risk factors—such as physical inactivity, smoking, risky alcohol consumption and obesity or overweight—and noncommunicable chronic diseases and disabilities (Loef and Walach 2012; Fine et al. 2004; Li et al. 2007; Pharr and Bungum 2012; Linardakis et al. 2015). The study focuses on working environment through health-related work benefits.

## 2.2. Workplace Health Promotion as an Internal CSR Activity

Researchers and practitioners interpret social responsibility in several different ways. In the literature, it is primarily identified with environmental awareness and environmental protection, however, it also has various other aspects today. Often, companies are not familiar with all aspects of this relatively new field, however, there is a growing tendency and willingness to try to find and apply CSR in practice. Social responsibility is undoubtedly one of the outstanding concepts of our economy today, which is briefly about how individuals, nations, and different companies can behave responsibly during their activities. Social responsibility and sustainability very often occur together in academic research, as in this situation the organization's traditional, short-term market-oriented interest is pushed into the background and other longer-term plans come to the fore even if its interest may not be directly measurable (Dos 2017). Moreover, Kot and Brzezinski (2015) emphasized in their research that a well-structured, organized and implemented, strongly enforced policy is crucial to facilitate the sustainable development. Furthermore, Grabara et al. (2016) stated that social responsibility itself has become a significant dimension of development at domestic and international level, in addition at micro and macroeconomic level. While most research agree that social responsibility is a strong business requirement, there is little consensus on what constitutes and how to implement it into corporate operations. The problem is that knowledge develops in parallel in different business disciplines, therefore opinions, ideas and feasibility intentions appear in many different ways. In addition, there are the cultural differences found in the global business environment (Kashyap et al. 2011). Similarly, Taras et al. (2011) stated that many researchers have verified that national culture values significantly influence the attitudes over the organizational culture.

The term of corporate social responsibility (CSR) has become dominant in the life of organizations nowadays, though there are also opponents of this concept (e.g., Chwistecka-Dudek 2016). Due to the many different definitions and approaches of CSR, it is often misunderstood and misinterpreted; or hardly understood its role in sustainable business models (Dahlsrud 2006; Doh et al. 2015; Ling 2019). The first formal definition derived from Bowen (1953) who stated that companies have a decision-making power which may have an impact on their actions and influence also the society as a whole. Although there

is not a unified definition of CSR, the most accepted and most frequently cited version is determined by the European Commission: "CSR is the responsibility of enterprises for their impact on society" (European Commission 2010). In order to meet their corporate social responsibility requirements, enterprises should have in place a process to integrate social, environmental, ethical and human rights concerns into their business operations and core strategy in close collaboration with their stakeholders (Macassa et al. 2017). This definition of CSR places the responsibility of enterprises on the three main above-mentioned pillars of sustainability, of which CSR and workplace health promotion are in the social pillar (Cochran 2007; Stawicka 2018). One of the leading paradigms of corporate social responsibility is Carroll's CSR Pyramid framework which determines four dimensions (economic, legal, ethical and philanthropic) (Carroll 1979; Carroll 2016). Friedman stated that CSR activities represent an unnecessary investment of shareholders, and that social responsibility should be the personal choice of individuals, not a business issue (Friedman 1970; Thao et al. 2019). Furthermore, Szegedi et al. (2020) developed a CSR index in accordance with stakeholder theory and examined the relationships between CSR and financial performance.

Based on the groups of stakeholders, two different areas of CSR activities can be distinguished—external activities related to external stakeholders (consumers, competitors, government and suppliers), and internal activities related to internal stakeholders (employees and other shareholders) as Figure 2 illustrates.

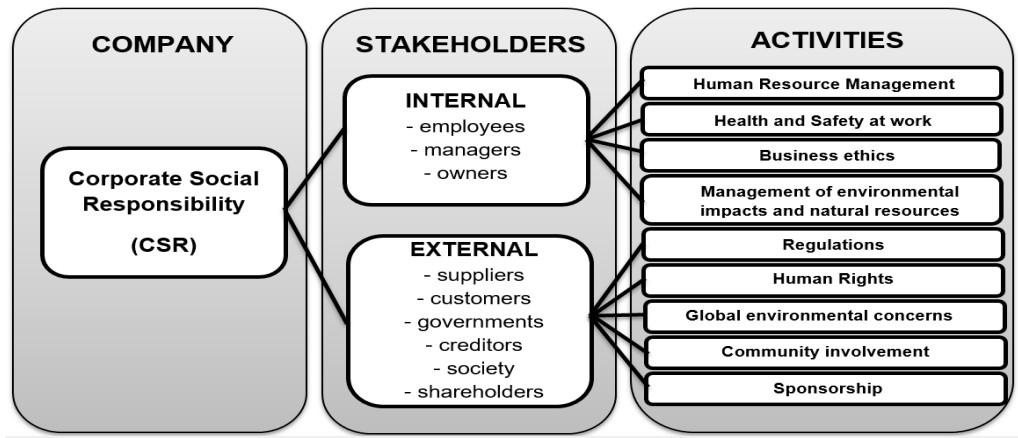

**Figure 2.** Areas of corporate social responsibility (CSR) activities based on company's stakeholders. Source: Authors' own edition based on Kerekes and Wetzker (2007) and Witek-Crabb (2019).

Despite the fact that most research still focuses on external CSR activities (Lichtenstein et al. 2004; Sen et al. 2006), some recent studies assess the impact of companies' CSR activities on employee attitudes and behaviors (Brammer et al. 2007; Turker 2009; Zhu et al. 2014). Studies found a significant, positive and long-term impact of CSR on community health, especially in developing countries and among socially excluded groups of population (Werner 2009). In Hungary, some companies already have started to introduce CSR activities, however it is mainly typical in life of multinational companies and even less easy and effective at the SME level.

The WHO has been working on occupational health since 1950, emphasizing that not only work and the workplace affect the physical and mental health of employees, but also the health of employees affects the company and its business performance (Szabo and Juhasz 2019). Based on the Ottawa Charter (WHO 1986), health promotion is "the process of enabling people to increase control over, and to improve, their health. To reach a state of complete physical, mental and social wellbeing, an individual or group must be able to identify and to realize aspirations, to satisfy needs, and to change or cope with the environment" (WHO 1986). If we talk about occupational health, it is necessary to properly determine the definition of workplace health promotion (WHP). The most

integrated and widely cited meaning is described by the European Network for Workplace Health Promotion (ENWHP) as a "modern corporate strategy which aims at preventing ill-health at work (including work-related diseases, accidents, injuries, occupational diseases and stress) and enhancing health-promoting potentials and wellbeing in the workforce" (ENWHP 2007). According to the ENWHP (2007), the most important areas where various measures and/or activities can be taken related to health promotion consist of lifestyles, ageing, corporate culture including leadership, development of employees, work-life balance, mental health and stress management, wellness, nutrition and physical health, corporate social responsibility (CSR). In contrast, few research show that CSR activities can motivate employees to initiate commitment, however, CSR performance does not result increasing job performance (Houghton et al. 2009; Vlachos et al. 2014).

Due to the market and profit-oriented approaches, it can be easily forgotten that corporate image and performance should not be evaluated only through financial performance or profit (Stojanovic et al. 2020; De Roeck et al. 2016). Kot (2014), for example, examined the research field by distinguishing five main areas of CSR business benefits: a positive effect on company image and reputation, a positive effect on employee motivation, retention and recruitment, cost saving, revenue increases from higher sales and market share, CSR-related risk reduction and management. Other research shows that healthier employees are already able to perform better physically and mentally in the short term, and they become more health-conscious, more efficient, more productive, more satisfied, more motivated and more loyal (Szabo and Juhasz 2019; Ozminkowski et al. 2016; Hendriksen et al. 2016; Gubler et al. 2017). In addition, Dumitrescu and Simionescu (2015) conducted empirical analyses based on accounting measures to determine company financial performance related to CSR. Besides increasing revenues, other financial benefits of CSR have been observed through costs of production and equity reduction (Matthiesen and Salzmann 2017). Moreover, Fehér and Reich (2020) verified in their research that workplace health management has a significant positive impact on the attractiveness of the workplace and employer. In addition, it may improve also the image of the company. Table 1 summarizes the most relevant benefits of WHP in short, middle and long term.

**Table 1.** Corporate benefits from workplace health promotion.

| In Few Months | After 1–2 Years | After 3–5 Years |
|---|---|---|
| Closer engagement | Higher productivity and performance | Less workplace injuries |
| Better workplace morale | Increased labor retention and attractiveness | Less disease and absence |
| Stronger team spirit, community building | Positive image | Less presenteeism |
| | Better individual health awareness | Improving returns on training and development |
| | Greater (increased) satisfaction | |

Source: Own edition based on Szabo and Juhasz (2019) and Tasmania (2012).

### 2.3. Employee Wellbeing, Satisfaction and Loyalty

As previous researchers have stated, the concept or issue of wellness is usually examined and discussed in terms of multiple dimensions. Most of them distinguish five–six dimensions (Roscoe 2009; Harari et al. 2005; Hettler 1984; Adams et al. 1997). One of the most well-known and cited methods to measure wellness is the Perceived Wellness Survey which include six following dimensions of wellness: physical, emotional, social, psychological, intellectual and spiritual (Adams et al. 1997). The criticism of this method is the excessive fragmentation of the psychological dimension into emotional, intellectual, psychological, social and spiritual parts; however, these concepts can be easy to confuse.

Similarly, Hettler's Wellness Hexagon consists of six different dimensions: physical, emotional, social, intellectual, spiritual and occupational. Wellbeing is a multidimensional and conceptually similar concept to wellness. For example, Hooker et al. (2021) examined eight dimensions of wellbeing in their model. Based on Linton et al. (2016) wellbeing includes several dimensions related to mental wellbeing (happiness and emotional quality of life), social wellbeing (social relationships and communities), spiritual wellbeing, activities and functioning (having activities to fill one's time), physical wellbeing (quality of physical performance and functioning); and personal circumstances (environmental and socioeconomic pressures and concerns). Summarizing the related literature sources physical health can be connected to the current psychical status of the people, mental health can be defined through the cognitive abilities and mental confusion—it is determined by various biological, environmental and socioeconomic factors. Social and emotional health is closely linked to the wellbeing and happiness arising from recognition, social relationships and activities (WHO 2018; Soo You and Lee 2006). The following model with eight dimensions of wellbeing—illustrated by Figure 3—can be divided by internal and external factors and includes both personal satisfaction (as spiritual harmony) and employee satisfaction (as occupational harmony).

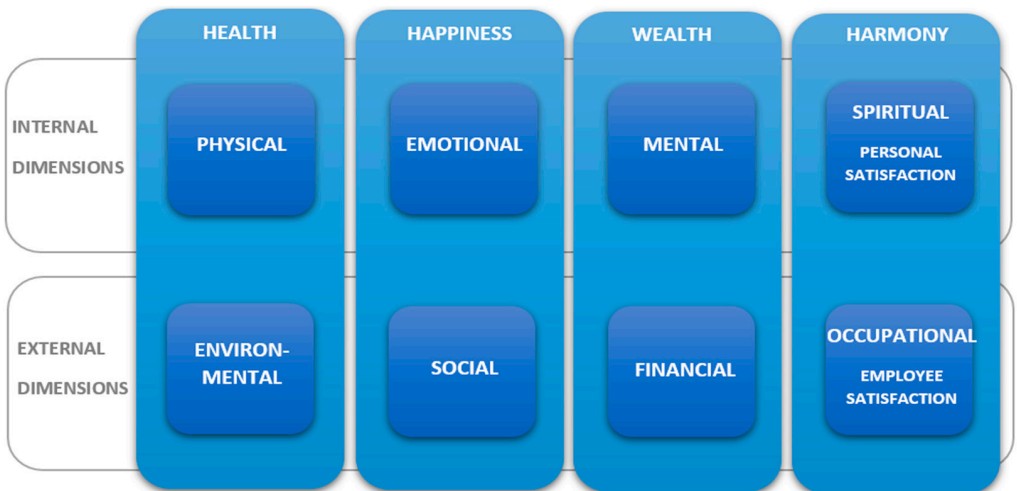

**Figure 3.** Dimensions of wellbeing. Source: Authors' own compilation based on Roscoe (2009); Meiselman (2016) and Adams et al. (1997).

Knowledge economy and knowledge-based society have become more important phenomenon today and therefore employee satisfaction and loyalty have become critical issues (Matzler et al. 2003; Renzl 2003). Employee motivation and retention are important determinants here as in the case of lean thinking and implementation of lean management system (Olah et al. 2017). There is no doubt that employee satisfaction is an important variable which is able to provide a general view about general emotion and feelings of employees about their job and workplace. However, it is difficult to measure due to its latent determinants. Employee satisfaction is measured and evaluated in several empirical studies either as an overall feeling about the work itself, or as a related set of attitudes about various aspects of the work (Spector 1997). Some researchers started to observe this area at the beginning of the last century. Taylor (1911), for example, recognized the economic importance of employee satisfaction, that employees work harder for cash rewards and higher pay. Locke (1976) have collected nearly 3000 studies that address some aspect of employee satisfaction. These surveys are mostly about the measure of the relationships between satisfaction and some other factor such as task management, leadership, reward system, group processes and so on. According to Locke (1976), employee satisfaction is a pleasant, positive feeling that results from an appreciation of work-related experiences which definition is closely correlated to Tayler's original concept. According to Hoppock (1935) employee satisfaction can be determined as a mix of cognitive, physiologic and

environmental factors that make the employees satisfied or dissatisfied with their work. Once employees are satisfied with their work, they will have a positive attitude towards it, and this is true in the opposite direction (Armstrong 2006). Herzberg's (1966) two-factor theory distinguishes between job characteristics: satisfaction can be caused by internal (motivating) factors such as responsibility, promotion, development opportunity, recognition, content and significance of the task; and external (so-called hygiene) factors not directly related to work: corporate policy and administration, management style, working conditions, personal relationships with the manager, employees and subordinates, position, job security, salary, remuneration system (Tietjen and Myers 1998).

Several empirical studies have found strong relationships between employee satisfaction, organizational commitment and loyalty (e.g., Mak and Sockel 2001; Martensen and Gronholdt 2001). Some of them also stated that employee satisfaction is negatively related to turnover (e.g., Tekleab et al. 2005; Ward 1988) and absenteeism (e.g., Muchinsky 1977). By creating joint values and socially responsible engagement enterprises can gain such competitive advantages as increasing competitiveness, image building or a satisfied and loyal workforce (Shpak et al. 2018). So called fringe benefit is suck a work benefit as the material and nonmaterial incentives the company offer and provide to its employees to commit them to the company. Employees received these benefits in addition to their wages or salaries, in some cases also after retirement (Khuong and Tien 2013). According to the result of Artz (2010) work benefits have significant and positive impact on employee satisfaction, and it is likely to have a profound impact on employee loyalty. It plays as a motivator factor helping to improve employee performance and to reduce employee turnover (Kasper et al. 2012).

Empirical studies started to deal with also the effects of COVID-19 pandemic on CSR activities, workplaces and work itself. Health sensitivity has significantly increased; important health factors have to be emphasized by not only policy decision makers but also employers. The workplaces can be redesigned, and work can be reimagined in response to the current health challenge. For example, currently empty workplaces can be changed creating and building opportunities for standing desks, healthy snacks or workplace activity/exercise programs. Furthermore, it can be clearly seen that companies especially from legal, financial or technology sectors—where employees can work from home—remain as productive and competitive with the transition to home office (Duffy et al. 2021; Zhang et al. 2021; O'Brien et al. 2021). Table 2 summarizes all health-related work benefits (as independent variables) which may have significant effects on employee wellbeing, satisfaction and through of them on loyalty (as dependent variables) based on our assumption. Table 2 includes also the relevant scientific literature sources related to each variable and offer a brief description about them.

Based on the above-discussed literature sources, and previous studies performed by Authors, indicators of exogenous variables and items connected to employee wellbeing are newly established and tested through exploratory factor analysis. Indicators related to employee satisfaction and loyalty are adapted from Homburg and Stock (2000, 2004), respectively. The following hypotheses were formulated for testing in this research by using the PLS—structural equation modelling (SEM) path modelling. A corresponding research framework highlighting the path and hypotheses are presented in Figure 4.

**Table 2.** Literature sources and empirical studies related to exogenous and endogenous variables.

| Variables | Factors/Constructs | Short Description of Indicators/Measures as Potential Work Benefits | Literature Sources Related to Factors |
|---|---|---|---|
| Exogenous variables | Physical health | Promoting sport activities, office exercises, corporate sport events, bike-sharing program organizing sport classes, providing suitable physical working conditions. | Turkyilmaz et al. (2011); Waqas et al. (2014); Roscoe (2009); Aazami et al. (2015); Brown et al. (2011) |
| | Emotional and social health | Promoting work/life balance and appreciation/recognition; prohibiting discrimination, bullying, harassment; supporting flexible working hours and home office. | Bataineh (2019); Rahman and Haleem (2018); Turkyilmaz et al. (2011); Strenitzerová and Achimský (2019); Waqas et al. (2014); Roscoe (2009); Yaseen (2020); Rani et al. (2011); Giovanis (2019); Han et al. (2021) |
| | Mental health | Stress management, organizing psychological counseling, coaching, mediation, relaxation, office massage, becoming family friendly workplace. | Aazami et al. (2015); Lee et al. (2009); Mansoor et al. (2011); Duraisingam et al. (2009); Mcdaid et al. (2009); Khuong and Linh (2020); O'Brien et al. (2021); Han et al. (2021) |
| | Healthy nutrition | Opportunity for fresh vegetable and fruit consumption, water filtration and fresh water, organizing healthy nutrition cooking courses and dietary counselling. | Turen et al. (2017); Andersen et al. (2017); Proper and van Mechelen (2007); Maldoy et al. (2021) |
| | Preventive care | Providing screening tests, vaccination, first aid trainings, supporting cessation programs. | Arocena et al. (2008); Warner et al. (2004); Smedslund et al. (2004); Warner et al. (2004); Asfar et al. (2019) |
| | Healthcare support | Providing health fund contribution, supporting recovery, regenerative holiday. | Skagen and Collins (2016); Kuoppala et al. (2008) |
| | Insurance | Providing financial support in case of illness, accident or death through health insurance. | Sears et al. (2014); O'Brien (2003) |
| Endogenous variables | Employee wellbeing | Positive feeling of employees related to their workplace, employer and work itself. | Roscoe (2009); Linton et al. (2016); Adams et al. (1997); Meiselman (2016); Baptiste (2008); Krekel Christian et al. (2019) |
| | Employee satisfaction | Satisfaction of employees with their work, workplace, employer; they like their job and do not intend to work for a different company. | Homburg and Stock (2004); Matzler and Renzl (2006); Turkyilmaz et al. (2011); Strenitzerová and Achimský (2019); Hassan et al. (2013); Khuong and Linh (2020); Rani et al. (2011); Giovanis (2019) |
| | Employee loyalty | Employees speak positively about their company, recommend their products/services; and would like to stay there in long term. | Homburg and Stock (2000); Matzler and Renzl (2006); Turkyilmaz et al. (2011); Strenitzerová and Achimský (2019); Murali et al. (2017); Hassan et al. (2013); Khuong and Linh (2020); Khuong et al. (2020); Giovanis (2019) |

Source: Authors' own edition.

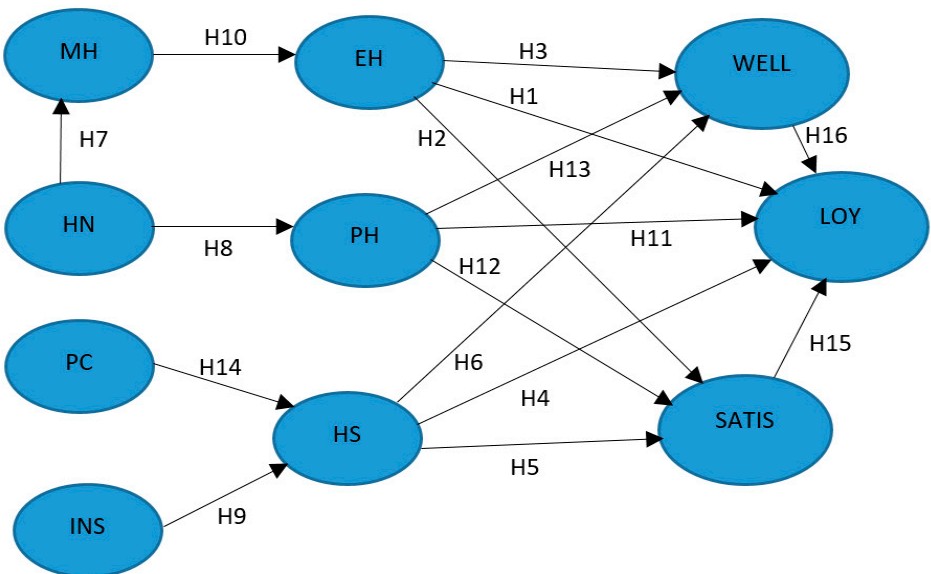

**Figure 4.** Research framework with corresponding hypotheses. Source: Authors' own edition.

The hypotheses are presented as below:

**Hypothesis 1 (H1).** *There is a positive impact of emotional health towards employee loyalty.*

**Hypothesis 2 (H2).** *There is a positive impact of emotional health towards employee satisfaction.*

**Hypothesis 3 (H3).** *There is a positive impact of emotional health towards employee wellbeing.*

**Hypothesis 4 (H4).** *There is a positive impact of healthcare support towards employee loyalty.*

**Hypothesis 5 (H5).** *There is a positive impact of healthcare support towards employee satisfaction.*

**Hypothesis 6 (H6).** *There is a positive impact of healthcare support towards employee wellbeing.*

**Hypothesis 7 (H7).** *There is a positive impact of healthy nutrition towards employee mental health.*

**Hypothesis 8 (H8).** *There is a positive impact of healthy nutrition towards employee physical health.*

**Hypothesis 9 (H9).** *There is a positive impact of Insurance towards healthcare support.*

**Hypothesis 10 (H10).** *There is a positive impact of mental health towards employee emotional health.*

**Hypothesis 11 (H11).** *There is a positive impact of physical health towards employee loyalty.*

**Hypothesis 12 (H12).** *There is a positive impact of physical health towards employee satisfaction.*

**Hypothesis 13 (H13).** *There is a positive impact of physical health towards employee wellbeing.*

**Hypothesis 14 (H14).** *There is a positive impact of preventive care towards healthcare support.*

**Hypothesis 15 (H15).** *There is a positive impact of employee satisfaction towards employee loyalty.*

**Hypothesis 16 (H16).** *There is a positive impact of employee wellbeing towards employee loyalty.*

## 3. Methodology

The main approach for this research is quantitative or structured approach. The design of this study follows survey research framework, and it is conducted by using a questionnaire. In order to design the questionnaire, different related scientific research articles and other literature sources were analyzed systematically. Based on literature review, content analysis of related CSR reports, 2 focus groups and exploratory in-depth interviews, above-mentioned hypotheses were formulated via the conceptual framework.

Due to the pandemic situation questionnaire survey was completely online and it was distributed via e-mails and also shared in social media. For data collection judgmental and snowball sampling technique was used. After data collection 537 completely filled surveys were received back for data analysis.

The structure of the questionnaire survey was the following. The first section of the questionnaire contains 8 questions and deals with current consumer habits and behaviors, the current health status and level of health-conscious behavior of the respondents. It was followed by the second major section including 5 questions which is related to social media usage habits. Since almost every multinational company and corporation today has a social media interface where they can also reach their own employees, Authors would like to explore whether social media can be a suitable tool to improve the relationship with employees, and to develop their health awareness. In third section the health-related work benefits are detected and measured by using five-point Likert scales. These 47 questions served as indicator questions for the factors of the structural model. The final section consists of 6 questions was related to demographic characteristics in order to present the sample. The target population of the study were adult employees with relevant work experience, the sampling location was in Hungary. The period of data collection was between December 2020 and January 2021.

Based on the available database from 537 respondents, data analysis was performed by using the partial least square (PLS-SEM) method, which is a statistical method that combines factor analysis, correlation and regression analysis to analyze the collected data. Literature sources suggest using the method of sample size is too small, applications do not have available theory, predictive accuracy is paramount and/or correct model specification cannot be ensured (Bacon 1999; Hwang et al. 2010; Wong 2013). The greatest advantage of this method is that also indirect effects can be examined in addition to the direct effects between the variables. Therefore, it was possible to observe and analyze how variables exert their effect on the target variables through other (mediator) variables. SEM was modelled with SmartPLS version 3.2.8 software (SMARTPLS GMBH 2019; Sarstedt et al. 2014; Sarstedt et al. 2011; Ringle et al. 2013; Nathan et al. 2019; Victor et al. 2019; Gonda et al. 2020). Furthermore, Statistical Package for Social Sciences (SPSS) version 21 was used to perform descriptive statistical analysis.

## 4. Results and Discussion

This section presents the result of the empirical study by firstly presenting the descriptive statistics, followed by the hypothesis testing results and path model result of PLS-SEM.

### 4.1. Descriptive Statistics for Demographic Characteristics of Respondents

Before describing the demographic characteristics, it is essential to notice that respondents cannot be characterized by a representative sample with national coverage. Respondents must be adults and they must have relevant work experience—no other inclusion and exclusion criteria were set up during data collection. Most respondents were women, with exactly 338 women (62.9%) and 199 men (37.1%) completing the questionnaire. The vast majority of respondents were in age groups of 30–39 (32.4%) and 40–49 (37.8%). This can be evaluated as a normal distribution, regarding the rate of these two groups in active population. The largest proportion of the respondents have higher education level (27.7% of them have BSc degree, more than 40% of them have MSc degree and 8.4% of them

have postgraduate, doctoral degree), which may determine and correlate with their income status. Based on their self-evaluation the net income per capita in their family is above the average (50.8%) or much higher than average (17.1%) in Hungary. In terms of industry where respondents work, they represent almost every group in a similar proportion.

### 4.2. Factors and Indicators in the Structural Model

Independent variables where factor indicators can be used to measure:

Factor 1: physical health (PH)
Factor 2: mental health (MH))
Factor 3: healthy nutrition (HN)
Factor 4: preventive care (PC)
Factor 5: healthcare support (HS)
Factor 6: insurance (INS)
Factor 7: emotional health (EH)

Dependent latent variables can be seen in the following list:

Factor 8: employee wellbeing (WELL)
Factor 9: employee satisfaction (SATIS)
Factor 10: employee loyalty (LOY)

According to the assumption of the authors (deriving from the descriptive statistics of independent and dependent variables), the variables should be evaluated in three layers—the first layer variables have an impact on second layer and subsequently the third layer. Different categories of variables can be seen in Table 3.

**Table 3.** Structure of variables.

| First Layer | Second Layer | Third Layer |
|---|---|---|
| Insurance | Healthcare support | Employee wellbeing |
| Preventive care | Physical health | Employee satisfaction |
| Healthy nutrition | Emotional health | Employee loyalty |
| Mental health | | |

Source: Authors' own edition.

### 4.3. PLS-SEM—Measurement Model Results

Data analysis should be started by assessing the construct validity and consistency reliability of the measurement model (Hair et al. 2016). According to the rule defined by Hair et al. (2016), all the outer loadings should above the threshold value of 0.70 to measure the individual item reliability and composite reliability (CR) should be higher than 0.7 thresholds (0.60 to 0.70 is considered acceptable) to measure the construct internal consistency in PLS (Khuong and Linh 2020). There were some indicators with outer loadings' Cronbach Alpha values under 0.7, therefore, all of them were removed from final model. Overall, five items were removed from scale measurement—one item from healthy nutrition, three items from mental health and one item from physical health were eliminated from the scale. Construct validity is determined by convergent validity and discriminant validity. Average variance extracted (AVE) is used to examine convergent validity. According to Fornell and Larcker (1981), AVE values should be above 0.5. AVE values of all indicators exceeded 0.5 and the composite reliability of the factor model was higher than 0.7 in all cases. AVE value indicated that constructs achieve adequate convergent validity. Table 4 demonstrates all values of composite reliability and convergent validity related to the model.

**Table 4.** Reliability and convergent validity.

|  | Cronbach's Alpha | rho_A | Composite Reliability (CR) | Average Variance Extracted (AVE) |
|---|---|---|---|---|
| **Emotional health** | 0.881 | 0.884 | 0.913 | 0.680 |
| **Healthcare support** | 0.841 | 0.847 | 0.894 | 0.680 |
| **Healthy nutrition** | 0.805 | 0.809 | 0.885 | 0.720 |
| **Insurance** | 0.891 | 0.907 | 0.917 | 0.649 |
| **Employee loyalty** | 0.893 | 0.903 | 0.926 | 0.758 |
| **Mental health** | 0.787 | 0.787 | 0.864 | 0.615 |
| **Physical health** | 0.759 | 0.758 | 0.847 | 0.582 |
| **Preventive care** | 0.737 | 0.765 | 0.834 | 0.558 |
| **Employee satisfaction** | 0.902 | 0.91 | 0.925 | 0.672 |
| **Employee wellbeing** | 0.769 | 0.777 | 0.867 | 0.685 |

Source: Authors' own calculation.

Multicollinearity analysis was performed using the Heterotrait-Monotrait ratio of correlations (HTMT) criteria. It is recommended that HTMT values should be ideally be below 0.85 (Henseler et al. 2015). All the values of HTMT in Table 5 achieve this, with the exception of employee satisfaction and employee loyalty where there HTMT value is 0.941. This is due to the close similarity between the item measures for both variables, although they are not identical. Moreover, both these variable items are adopted from previous empirical studies which validates the criterion validity. Additionally, referring to (Hair et al. 2017), HTMT values above 0.9 is not desirable while 0.95 is the threshold that will make it definitely undesirable. Hence the value in this result is below 0.95, while taking into account these variables (satisfaction and loyalty) were using items previously tested and validated in former empirical research and having met composite reliability high scores and passed the AVE threshold test, this study retains the variables for hypothesis testing.

**Table 5.** The Heterotrait-Monotrait ratio of correlations (HTMT) result.

|  | EH | HS | HN | INS | LOY | MH | PH | PC | SATIS |
|---|---|---|---|---|---|---|---|---|---|
| Emotional health |  |  |  |  |  |  |  |  |  |
| Healthcare support | 0.398 |  |  |  |  |  |  |  |  |
| Healthy nutrition | 0.237 | 0.489 |  |  |  |  |  |  |  |
| Insurance | 0.465 | 0.742 | 0.448 |  |  |  |  |  |  |
| Employee loyalty | 0.112 | 0.065 | 0.05 | 0.047 |  |  |  |  |  |
| Mental health | 0.392 | 0.417 | 0.670 | 0.405 | 0.055 |  |  |  |  |
| Physical health | 0.185 | 0.409 | 0.59 | 0.339 | 0.090 | 0.564 |  |  |  |
| Preventive care | 0.289 | 0.610 | 0.673 | 0.645 | 0.041 | 0.547 | 0.560 |  |  |
| Employee Satisfaction | 0.127 | 0.058 | 0.06 | 0.056 | 0.941 | 0.051 | 0.138 | 0.045 |  |
| Employee wellbeing | 0.78 | 0.461 | 0.407 | 0.576 | 0.137 | 0.496 | 0.456 | 0.515 | 0.133 |

Source: Authors' own calculation.

### 4.4. PLS-SEM—Structural Model Results

This section presents the results of the PLS path model analysis which is used to test the research hypotheses. Table 6 presents the path coefficient results, t-stats and r-square values for endogenous factors.

Based on the result, this study supports H3, H5, H6, H7, H8, H9, H10, H12, H13, H14 and H15 (eleven hypotheses supported); while H1, H2, H4, H11 and H16 (five hypotheses) are not supported. Based on the findings, Figure 5 is formed as below to highlight the significant paths based on the supported hypotheses which depicts the impact of the research independent variables (layer 1 and layer 2 variables) towards the dependent variables (layer 3 variables).

**Table 6.** Results of PLS Path Modelling and Hypotheses Testing.

| Hyp. | Relationship | Path Coef. | t-Stats | *p*-Value | r-Square |
|:---:|:---:|:---:|:---:|:---:|:---:|
| H1 | Emotional health -> loyalty | −0.011 | 0.311 | 0.756 | 0.737 |
| H2 | Emotional health -> satisfaction | 0.101 | 1.557 | 0.120 | 0.034 |
| H3 | Emotional health -> wellbeing | 0.584 | 9.405 | 0.000 * | 0.488 |
| H4 | Healthcare support -> loyalty | −0.016 | 0.61 | 0.542 | |
| H5 | Healthcare support -> satisfaction | −0.139 | 2.666 | 0.008 * | |
| H6 | Healthcare support -> wellbeing | 0.1 | 2.044 | 0.042 * | |
| H7 | Healthy nutrition -> mental health | 0.535 | 15.853 | 0.000 * | 0.287 |
| H8 | Healthy nutrition -> physical health | 0.461 | 11.44 | 0.000 * | 0.212 |
| H9 | Insurance -> healthcare support | 0.551 | 11.98 | 0.000 * | 0.458 |
| H10 | Mental health -> emotional health | 0.327 | 7.857 | 0.000 * | 0.107 |
| H11 | Physical health -> loyalty | −0.035 | 1.629 | 0.104 | |
| H12 | Physical health -> satisfaction | 0.117 | 2.611 | 0.009 * | |
| H13 | Physical health -> wellbeing | 0.215 | 5.066 | 0.000 * | |
| H14 | Preventive care -> healthcare support | 0.198 | 3.937 | 0.000 * | |
| H15 | Satisfaction -> loyalty | 0.858 | 53.416 | 0.000 * | |
| H16 | Wellbeing -> loyalty | 0.039 | 1.046 | 0.296 | |

Note: * path is significant at *p*-value below 0.05. Source: Authors' own calculation.

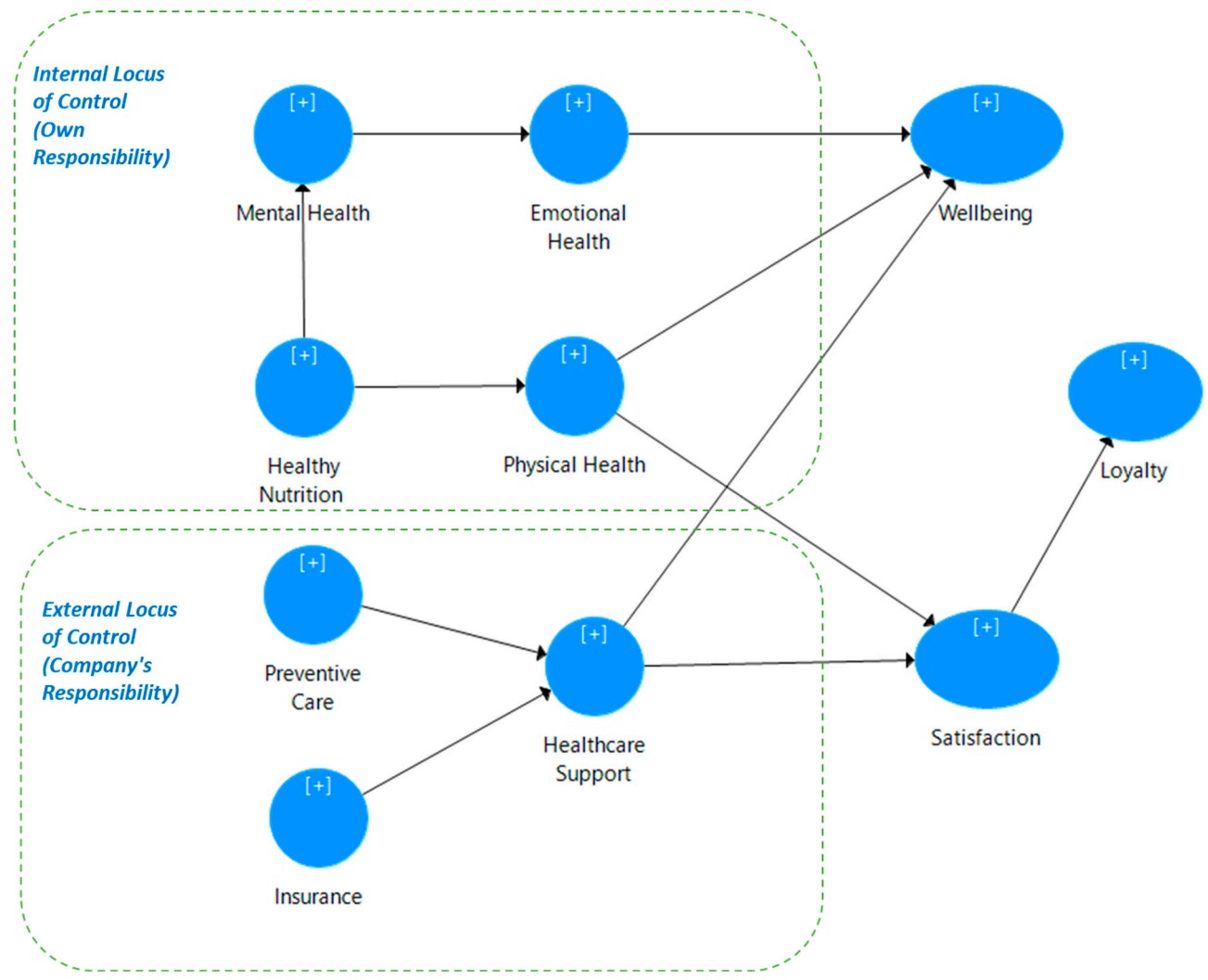

**Figure 5.** Relationships among variables and direct effects in the model. Source: Authors' own edition.

The workplace employee wellbeing, satisfaction and loyalty dynamics from the perspective of employees seem to be explained in 3 major dimensions. They seem to be centered around three major categories of determinants that impacts wellbeing, satisfaction and loyalty uniquely. This study finds, (1) healthcare support (2) physical health and (3) emotional health are important pre-determinants to employee wellbeing (Figure 5).

Healthcare support (HS) seem to be perceived as the "external locus of control" here, where facilities such as insurance and preventive care (tangible facilities) are perhaps perceived as purely at the control and responsibility of employer. Both are very important to complete a functional HS system of an organization. HS is also an important determination of employee wellbeing and satisfaction at workplace. However, it does not directly lead to Loyalty of employees to workplace, probably because it is viewed as "Hygiene Factor" and not a motivating factor for employees.

Healthy nutrition (HL) shows a strong impact towards physical health (PH) and also towards mental health (MH). Therefore, having HN is understood by participants as vital to support their PH and MH; and these (PH and MH) are perhaps seen by participants as within their "internal locus of control", which they have direct control. MH further strengthens their emotional health (EH) and eventually their wellbeing. These factors that are perceived as within employees' own "internal locus of control" strongly predicts their wellbeing, but do not impact their Loyalty to their workplace. Employees feel that these "internal locus control" factors such as having healthy nutrition, managing own mental and physical health as well as their emotional health, as perhaps "own responsibility" and since they are responsible for them, so too is their wellbeing at the workplace.

The findings seem to suggest the invisible layers of "internal locus of control" and "external locus of control" in this study. The internal locus of control factors i.e., HN, MH, PH and EH are self-managed and hence these factors lead to their wellbeing, but do not lead to workplace satisfaction and loyalty. On the contrary, facilities that are provided by the organization "external locus of control" i.e., PC, INS and HS have a twin-role in impacting employee wellbeing and satisfaction with their workplace. It shows that fundamental HC support systems such as PC and INS must be in place in a workplace to ensure employee satisfaction and eventually their loyalty. These are considered hygiene factors. Employee wellbeing appears as a standalone endogenous construct, which is largely impacted by employee "self-initiatives factors", hence although organizations may be advocating "wellbeing" as an organization-driven initiative at the workplace, employees still perceive wellbeing as their own locus of control and do not attribute it towards their satisfaction and loyalty towards the organization. This is a peculiar finding in this study as previous literatures in organization development have often supported strong linkages between organizational initiative to employee wellbeing to their productivity and retention.

Based on these findings, we theorize that during times of crisis, employees become more self-reliant and think about self-preservation. During this process, perhaps they disassociate the promotion of wellbeing at workplace and their loyalty to workplace. Especially during this recent COVID-19 pandemic, most employees were working from home and did not have physical presence in the office or work premises. As such, they would have felt less physical protection from workplace and hence resort to self preservation and reliance in order to cope with the new norm. As this is a new phenomenon observed through the findings of this empirical research, we theorize this behavior as workers' self-reliance and preservation. It is a condition where workers become more conscious of their own role and become self-dependent for their own wellbeing. Although employee wellbeing efforts are also promoted by their employer, employees seem to view it as their internal locus of control during pandemic. This can be viewed as an act of self-preservation and survival during times of crisis and pandemic.

*4.5. Limitations and Directions for Future Study*

Despite this research provides a better and broader understanding of the impacts of health-related work benefits (and thus, workplace health promotion) on employee

wellbeing, satisfactions and loyalty, it also has several limitations listed as followings. The survey was conducted in Hungary. The empirical study is based on the results derived from this country, and respondents cannot be characterized by a representative sample with national coverage. Nevertheless, PLS-SEM is suitable to observe and examine the impact of latent variables in smaller sample as well. In addition, the validity of conclusions need not be restricted only to Hungary, since theoretical and empirical results have relevance also in international environment.

Due to the current COVID-19 pandemic situation it would be worth and useful to conduct a repeated research with a much larger sample size and deeper diversification in order to examine if physical and mental health are considered to be such important factors also after pandemic. Due to this situation, in these days, employees cannot take advantage of certain health related work benefits. Furthermore, some benefits can seem like compulsion now, not a real work benefit (for example home office).

This study could not use probability sampling methodology due to difficulty in obtaining reliable sampling frame of all employees from all sectors in Hungary during the time of the data collection. This could be an agenda for future research in this area of research. Furthermore, the research is planned to extend internationally and make comparative analysis between countries. It would be necessary to test the self-reliance and Preservation theory in workplace settings of other cultures and country. Future study could also harness the power of social media data to capture employees' habits and behaviors related to the development of their health awareness and its impact to their wellbeing and loyalty to workplace.

## 5. Conclusions

The research paper explains the different health-related work benefits as key factors which play an important role not only in employee wellbeing but also in employee satisfaction and loyalty. Studying the relevant literature sources, it can be clearly seen that there is not a comprehensive study on these specific factors. The knowledge of non-financial motivating factors is crucial for employers, especially nowadays, when health is more appreciated by the employees. This research discovers new relationships among employee workplace wellbeing, satisfaction and loyalty variables. Data analysis derived from responses of large number of employees in Hungary reveal new insights to explain employee wellbeing at workplace. Based on the results, eemployee wellbeing, satisfaction and loyalty dynamics from employees' point of view seem to be explained and evaluated in three major dimensions: physical health, emotional health and healthcare support which led to employee wellbeing. The importance of physical health, mental health and thus emotional health has increased likely due to the COVID-19 pandemic. We found disassociation between wellbeing and loyalty which is peculiar, and we explain this by theorising workers self-reliance and preservation behaviour during times of crisis.

Nowadays with home office arrangement, flexible working hours and work/life balance are highly appreciated by the employees, therefore emotional health has become especially important. However, the respondents considered these factors as their own responsibilities and hence physical and emotional health lead to wellbeing but do not affect neither their satisfaction nor their loyalty at workplace. This outcome could be a reason due to the pandemic situation. In contrast, external locus of control variables or factors such as healthcare support can be considered as important determination of employee wellbeing and satisfaction. However, it does not directly cause loyalty, probably because it is viewed as "Hygiene Factor" and not a motivating factor for employees. Further findings highlight that health sensitivity has increased since the willingness to respond was significantly above normal. This finding shows that health concerns arouse greater interest among employees during the pandemic.

**Author Contributions:** Conceptualization, E.G.-H.; methodology, E.G.-H. and R.J.N.; software, E.G.-H. and R.J.N.; validation, E.G.-H. and R.J.N.; formal analysis, E.G.-H. and R.J.N.; investigation, E.G.-H.; resources, E.G.-H., R.J.N. and M.F.-F.; data curation, E.G.-H. and R.J.N.; writing—original

draft preparation, E.G.-H.; writing—review and editing, E.G.-H., R.J.N. and M.F.-F.; visualization, E.G.-H.; supervision, M.F.-F.; project administration, E.G.-H.; funding acquisition, M.F.-F. All the authors discussed the results, and implications and commented on the manuscript at all stages. All authors have read and agreed to the published version of the manuscript.

**Funding:** This research received no external funding.

**Informed Consent Statement:** Not applicable.

**Data Availability Statement:** Not applicable.

**Conflicts of Interest:** The authors declare no conflict of interest.

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
