# Peer review of "Workplace Health Promotion, Employee Wellbeing and Loyalty during Covid-19 Pandemic—Large Scale Empirical Evidence from Hungary"

_economies, doi:10.3390/economies9020055_

Round 1

Reviewer 1 Report

The paper presents in my view an actual and intersting study. The literature review gives a considerable ammount of information and thus a solid background for the reader. The study design is plausible, the selected constructs make IMHO perfectly sense. My only complaint here is that it remains somewhat unclear how data with respect to the different constructs are collected (new or established items...?). The validity is tested by analysis of correlations, however, I feel there should be also a consideration about the content-related validity of the items. The presentation of the results and the conclusion are exemplarily clear.

The paper is very readable, but (searching hard...) I found one missing komma: after satisfaction in l. 68.

Author Response

Dear Chief Editor of Economies MDPI,

We - the Authors of this manuscript - would like to sincerely thank you and the reviewers of our manuscript for giving us constructive comments and suggestions to further improve this article.

Based on the comments and suggestions, we have carried out all necessary corrections and provided our responses to reviewers’ valuable insights and comments.

We thank you, the team at the editorial office and the reviewers once again for the time given to process our work and for the feedback that has surely improved our final manuscript and communication to the readers of Economies Journal. Detailed responses to the reviewer’s comments and feedback can be found in the attachment.

Thank you.

Yours Sincerely,

The Authors.

Reviewer 2 Report

The manuscript aims to investigate how certain types of health-related work benefits contribute to employee wellbeing, satisfaction, and loyalty. Overall, the implemented statistical methodology is robust, and the internal validity seems high. The model is scientifically sound, and the evidence seems to support the conclusions. The topics treated are relevant and may be useful even to a broader public. Altogether, the paper is worthy of publication. Minor improvements are, however, possible (and therefore needed).

Title

The title does not seem to properly capture the "geist" of the paper, which is, according to the abstract, the introduction of CSR as an important building block for social sustainability and employees' wellbeing. Moreover, the title emphasizes the Covid-19 pandemic, but not much attention is devoted to the issue in the text. Indeed, the strongest link between Covid-19 and the model proposed in the manuscript seems to be that the data gathering was performed during the pandemic. The authors should elaborate more on the connections between Covid-19 (or pandemics in general) and the other concepts discussed within the text.

Introduction is lacking important citations

  1. The 3-levels taxonomy of health awareness as social sustainability factor seems to have good face validity; however, no papers or other relevant literature sources are cited to back it up.

  1. The fact that external CSR activities are better researched than internal ones (given their fundamental epistemic relevance for the study) needs to be backed up by some literature sources. Otherwise, it seems more  a claim than a statement.

  1. Extensive literature review, secondary research and expert consultations are mentioned (lines 69 - 70). Still, it is not clear from the sentence's wording whether they refer to the manuscript itself or others' work. In the latter case, it all should be cited.

Model-related issues

  1. It should be better explained on which criteria the indirect variables are based. Although the literature review presents different ways to measure satisfaction and wellbeing, at any point in the manuscript there is a proper explanation why exactly those seven variables were chosen as indirect. Also, I might be wrong, but it seems that the distinction between indirect and direct variables is not a part of the typical PLS-PM vocabulary. There do exist direct and indirect effects among observed variables. Still, variables themselves are usually categorized as latent and manifest or endogenous and exogenous. If there exists a specific reason for choosing a direct/indirect dichotomy, it should be clearly stated.
  2. For the sake of higher reproducibility, it is better to report the questions used in the survey (especially the 47 hypotheses-related questions) in the Appendix. 
  3. Why were EH, PH, and HS treated as endogenous latent variables (second layer of the model) instead of being considered exogenous (like MH, HN, PC, and INS)? No proper explanation is given throughout the manuscript.
  4. As for the model, I understand why PLS PM (of which I am a big fan myself) was chosen. However, it would be better to provide readers with more information about the model and its pros and cons.
  5. The problems that may arise because of the non-probabilistic sampling should also be specified.
  6. The "descriptive statistics" part is short and seems irrelevant to the conclusions. If possible, it should be expanded by means of bar plots or similar plots/charts (after all, one picture is worth 1000 words).

Conclusions

             I particularly liked the interpretation of PLS PM results about external and internal loci of control. However, it would be better to elaborate on the managerial relevance of your findings. Why is it important? What practical benefits do your findings have? Also, albeit a stylistic choice, "limitations and directions for future studies" seems to be better suited as the last paragraph of the manuscript.

Clarity issues

The manuscript is written in good English, although some stylistic imperfections and obvious misspellings are present throughout the text. Without citing them one by one, I advise you to proofread your manuscript carefully once more to detect and correct them.

Author Response

(The authors gave the same response as above.)
